# ISTA-NAS: Efficient and Consistent Neural Architecture Search by Sparse Coding

**Yibo Yang**[1,2], **Hongyang Li**[2], **Shan You**[3], **Fei Wang**[3], **Chen Qian**[3], **Zhouchen Lin**[2,*]

[1]Center for Data Science, Academy for Advanced Interdisciplinary Studies, Peking University
[2]Key Laboratory of Machine Perception (MOE), School of EECS, Peking University
[3]SenseTime
{ibo, lhy_ustb, zlin}@pku.edu.cn, {youshan, wangfei, qianchen}@sensetime.com

## Abstract

Neural architecture search (NAS) aims to produce the optimal sparse solution from a high-dimensional space spanned by all candidate connections. Current gradient-based NAS methods commonly ignore the constraint of sparsity in the search phase, but project the optimized solution onto a sparse one by post-processing. As a result, the dense super-net for search is inefficient to train and has a gap with the projected architecture for evaluation. In this paper, we formulate neural architecture search as a sparse coding problem. We perform the differentiable search on a compressed lower-dimensional space that has the same validation loss as the original sparse solution space, and recover an architecture by solving the sparse coding problem. The differentiable search and architecture recovery are optimized in an alternate manner. By doing so, our network for search at each update satisfies the sparsity constraint and is efficient to train. In order to also eliminate the depth and width gap between the network in search and the target-net in evaluation, we further propose a method to search and evaluate in one stage under the target-net settings. When training finishes, architecture variables are absorbed into network weights. Thus we get the searched architecture and optimized parameters in a single run. In experiments, our two-stage method on CIFAR-10 requires only 0.05 GPU-day for search. Our one-stage method produces state-of-the-art performances on both CIFAR-10 and ImageNet at the cost of only evaluation time[1].

## 1 Introduction

Current NAS studies can be mainly categorized into reinforcement learning-based [1, 60, 58, 61, 4], evolution-based [41, 40, 29, 35, 14], Bayesian optimization-based [25, 59], and gradient-based methods [30, 48, 5, 53]. Most reinforcement learning and evolution-based algorithms suffer from huge computational cost, while gradient-based methods are simple to implement.

In gradient-based methods, an over-parameterized super-net is constructed to cover all candidate connections. Different architectures are sub-graphs of the super-net by weight sharing [39]. Thus the aim of search is to determine a sparse solution from the high-dimensional architecture space. Liu *et al.* propose a differentiable search framework, DARTS [30], by introducing a set of architecture variables jointly optimized with the network weights. After search, the architecture variables are projected to be sparse by only keeping the top-2 strongest connections for each intermediate node as the target-net for evaluation. This method is the basis of many follow-up studies [48, 5, 9, 49].

However, we note that the architecture variables do not satisfy the sparsity constraint during search. There is a gap between the dense super-net in search and the sparse target-net in evaluation (the term

---

[*]Corresponding author.
[1]Code address: https://github.com/iboing/ISTA-NAS.

"evaluation" in this paper denotes re-training on the full train set and inference on the test set). From the perspective of optimization, solutions should be constrained in the feasible region at each iteration, as adopted in the Projected Gradient Descent (PGD) [36] and proximal algorithms [37]. Otherwise, the converged solution may be far from the optimal one in the feasible region. It explains the fact that there is a poor correlation between the performances of super-net for search and target-net for evaluation in DARTS [48, 7, 50]. A high-performance super-net does not necessarily produce a good architecture after projection onto the sparsity constraint. Besides, the dense super-net covering all candidate connections is inefficient to train due to its huge computational and memory cost.

In some follow-up studies [48, 13, 47, 7], the Gumbel Softmax strategy [24, 33] is adopted to enforce sparsification. Nevertheless, the super-net still contains the whole graph which is different from the derived architecture in target-net and does not reduce the computational and memory consumption. In ProxylessNAS [5] and NASP [53], only sampled or projected connections are active for each edge to reduce the memory consumption and search cost. But the active paths of super-net still deviate from the target-net as there is no connection for some edges in the target-net. For single-path architecture search, DSNAS [21] proposes a one-stage method to eliminate the inconsistency between super-net and target-net brought by two-stage methods, while there is yet no single-stage solution to DARTS-based architecture where the dimension of candidate connections is relatively higher.

Inspired by the observation that architecture variables in DARTS should have a sparse structure that can be well-represented by a compact space, in this paper, we propose to formulate NAS as a sparse coding problem, named ISTA-NAS. We construct an equivalent compressed search space where each point corresponds to a sparse solution in the original space. We perform gradient-based search in the compressed space with the sparsity constraint inherently satisfied, and then recover a new architecture by the sparse coding problem, which can be efficiently solved by well-developed methods, such as the iterative shrinkage thresholding algorithm (ISTA) [12]. The differentiable search and architecture recovery are conducted in an alternate way, so at each update, the network for search is sparse and efficient to train. After convergence, there is no need of projection onto sparsity constraint by post-processing and the searched architecture is directly available for evaluation.

In previous studies [30, 48, 49], the super-net in search is dense, and thus has a smaller depth and width than the target-net setting due to limited memory. However, the number of cells and channels has a significant effect on the search result [50, 9]. In order to also eliminate these gaps between super-net in search and target-net in evaluation, we develop a one-stage framework where search and evaluation share the same super-net under the target-net settings, such as depth, width and batchsize. One-stage ISTA-NAS begins to search with all batch normalization (BN) [23] layers frozen. When a termination condition is satisfied, architecture variables cease to change and one sparse architecture is searched and fixed. BN layers are now trainable and other network weights continue to be optimized. After training, architecture variables are absorbed into the parameters of BN layers, and we get the searched architecture and all optimized parameters in a single run with only evaluation cost.

We list the contributions of this paper as follows:

- We formulate neural architecture search as a sparse coding problem and propose ISTA-NAS, which performs the differentiable search on an equivalent compressed space and recovers its corresponding sparse architecture in an alternate manner. The sparsity constraint is inherently satisfied at each update so the search is more efficient and consistent with evaluation.
- We further develop a one-stage ISTA-NAS that incorporates search and evaluation in a single framework under the target-net settings. The network for search has no gap in terms of depth, width and even training batchsize with the derived architecture for evaluation.
- In experiments, the two-stage ISTA-NAS finishes search on CIFAR-10 in only 0.05 GPU-day. And the one-stage version produces state-of-the-art performances in a single run on CIFAR-10 or directly on ImageNet at the cost of only evaluation time.

## 2 Related Work

Neural architecture search (NAS) has been proposed to spare the manual efforts in traditional networks [42, 43, 19, 22, 52, 51, 55] and benefit application tasks, such as object detection [38, 16, 45, 10], and semantic segmentation [8, 28]. Current gradient-based search methods widely adopt a two-stage strategy. In the first stage for search, the dense super-net covers all candidate connections and has to decrease the depth and width. As a result, it has a gap with the target-net for evaluation in the second stage, and the two stages correlate poorly. Several studies have been proposed for this problem.

**Reducing the gap.** To derive an architecture that is close to the one during search, Gumbel Softmax [48, 13, 7] and sparsity or entropy regularization [56, 15] are adopted to enforce sparse architecture variables. But the whole graph still needs to be stored and computed, which is inefficient to train for the dense super-net. Some studies use a sparse super-net by only keeping sampled or projected connections active [5, 53]. P-DARTS [9] reduces the depth gap by gradually dropping connections and increasing depths. Even if these studies improve the correlation between the two stages, their super-nets only approach to but do not strictly satisfy the sparsity constraint. A post-processing is still needed to derive the target-net. Our method differs from theses studies in that we keep the sparsity constraint inherently satisfied at each update, and do not need the projection as post-processing.

**One-stage NAS.** Some studies [3, 34, 21] have proposed one-stage search methods to circumvent the gap brought by two-stage methods. The architecture variables and network weights are simultaneously optimized. However, they are all proposed for the chain-based search space [18, 54, 47]. We show that ISTA-NAS also enables one-stage search. As a comparison, our method is developed for the cell-based search space in DARTS, which has a higher dimension because candidate connections for each intermediate node come from various operations of multiple previous nodes.

We note that a study [11] introduces compressed sensing into NAS. But the method is based on a meta-learning algorithm, instead of our differentiable architecture search. Besides, [27] tries to combine optimization algorithms with neural architecture design, instead of search.

## 3   ISTA-NAS

In this section, we first make a brief review of the differentiable architecture search and sparse coding. Then we build their relation and formulate NAS as a sparse coding problem. Finally, we introduce our two-stage and one-stage ISTA-NAS algorithms, respectively.

### 3.1   Preliminaries on Differentiable Neural Architecture Search

The search space of a cell in DARTS-based method is formed as a directed acyclic graph (DAG) consisting of $n$ ordered nodes $\{x_1, x_2, \cdots, x_n\}$ and their edges $\mathcal{E} = \{e^{(i,j)} | 1 \leq i < j \leq n\}$. Each edge has $K$ candidate operations from $\mathcal{O} = \{o_1, o_2, \cdots, o_K\}$, such as identity, max-pooling, and $3 \times 3$ separable convolution. With a binary variable $z_k^{(i,j)} \in \{0, 1\}$ to indicate whether the corresponding connection is active, we have the intermediate node $x_j$ as:

$$x_j = \sum_{i=1}^{j-1} \sum_{k=1}^{K} z_k^{(i,j)} o_k(x_i) = \mathbf{z}_j^T \mathbf{o}_j, \tag{1}$$

where $\mathbf{z}_j \in \{0, 1\}^{(j-1)K}$ and $\mathbf{o}_j \in \mathbb{R}^{(j-1)K}$ denote the vectors formed by $z_k^{(i,j)}$ and $o_k(x_i)$, respectively. The goal of NAS is to solve the following constrained bi-level optimization problem:

$$Z^* = \underset{Z}{\operatorname{argmin}} \, \mathcal{L}_{val}(\mathcal{N}(W^*, Z)), \tag{2}$$

$$W^* = \underset{W}{\operatorname{argmin}} \, \mathcal{L}_{train}(\mathcal{N}(W, Z)), \tag{3}$$

$$s.t. \quad \|\mathbf{z}_j\|_0 = s_j, \, 1 < j \leq n, \tag{4}$$

where $Z = \{\mathbf{z}_j\}_{j=2}^n$, $W$ is the weights of super-net $\mathcal{N}$, and $s_j$ denotes the sparseness of node $j$. Since the architecture variables $\mathbf{z}_j$ are binary and thus stubborn to optimize in a differentiable way, current studies adopt the continuous relaxation by $z_k^{(i,j)} = \exp(\alpha_k^{(i,j)}) / \sum_k \exp(\alpha_k^{(i,j)})$ and optimize $\alpha_k^{(i,j)}$ as trainable variables [30, 48, 9, 49]. In the search phase, the sparsity constraint Eq. (4) is not considered. $\alpha_k^{(i,j)}$ and $W$ are optimized in an alternate manner by solving Eq. (2) and Eq. (3), respectively. After search, $\mathbf{z}_j$ is projected onto the sparsity constraint Eq. (4) by only keeping the top-2 strongest dimensions for node $j$, *i.e.*, $s_j = 2, \forall \, 1 < j \leq n$.

It is shown that current studies widely ignore the sparsity constraint Eq. (4) during search, and the sparse architecture variables $\mathbf{z}_j$ are derived by post-processing. This may cause an invalid search process because there is few correlation between the optimal solutions inside and outside the feasible region. Constraints should be satisfied at each step of optimization [36, 37]. In addition, the dense super-net $\mathcal{N}(W, Z)$ covers all candidate connections in search. As a result, it is inefficient to train, and has to adopt a small depth, width and training batchsize due to limited memory, which introduces the gap of depth, width and batchsize with the target-net in evaluation.

## 3.2 Preliminaries on Sparse Coding

Sparse coding serves as the basis of many machine learning applications, such as image denoising and super-resolution (see [57] and references therein). It aims to recover a sparse signal $\mathbf{z} \in \mathbb{R}^n$ from its compressed observation $\mathbf{b} \in \mathbb{R}^m$:

$$\mathbf{b} = \mathbf{A}\mathbf{z} + \epsilon, \tag{5}$$

where $\epsilon \in \mathbb{R}^m$ is the additive noise, $\mathbf{A} \in \mathbb{R}^{m \times n}$ is an over-complete measurement matrix and we have $m < n$. The sparsity constraint of $\mathbf{z}$ is non-convex and challenging to deal with. A popular approach is to use the $\ell_1$-norm as the convex surrogate, known as the LASSO formulation [46]:

$$\min_{\mathbf{z}} \frac{1}{2} \|\mathbf{A}\mathbf{z} - \mathbf{b}\|_2^2 + \lambda \|\mathbf{z}\|_1, \tag{6}$$

where $\lambda$ is the regularization parameter. Many well-developed algorithms are proposed to solve the problem Eq. (6), such as the iterative shrinkage thresholding algorithm (ISTA) [12]:

$$\mathbf{z}^{(t+1)} = \eta_{\lambda/L} \left( \mathbf{z}^{(t)} - \frac{1}{L} \mathbf{A}^T \left( \mathbf{A}\mathbf{z} - \mathbf{b} \right) \right), \quad t = 0, 1, \cdots, \tag{7}$$

where $L$ is taken as the largest eigenvalue of $\mathbf{A}^T \mathbf{A}$, and $\eta_\theta$ is the component-wise soft-thresholding operator defined as: $\eta_\theta(x) = \text{sign}(x) \max(|x| - \theta, 0)$. ISTA is a proximal gradient method applied to Eq. (6) and has an $O(1/t)$ convergence rate with a proper $\lambda$ [2].

## 3.3 Formulate Differentiable NAS as Sparse Coding

We also relax the binary constraint of $\mathbf{z}_j$ and let $\mathbf{z}_j \in \Omega(\mathbf{z}_j) = \{\mathbf{z}_j \in \mathbb{R}^{(j-1)K} : \|\mathbf{z}_j\|_0 \leq s_j\}$, where $s_j$ is the sparseness in Eq. (4). It is hard for current methods to search $\mathbf{z}_j$ directly in $\Omega(\mathbf{z}_j)$. If there is an equivalent search process constructed on a compressed space $\Omega(\mathbf{b}_j) = \mathbb{R}^{m_j}$, where $m_j < (j-1)K$, we can perform the differentiable search on the lower-dimensional space without the sparsity constraint Eq. (4), and then recover a $\mathbf{z}_j$ in $\Omega(\mathbf{z}_j)$ by sparse coding. We have $\mathbf{b}_j$ by a measurement matrix $\mathbf{A}_j \in \mathbb{R}^{m_j \times (j-1)K}$ as $\mathbf{b}_j = \mathbf{A}_j \mathbf{z}_j$. Assuming that $\mathbf{E}_j$ is a residual matrix of $\mathbf{A}_j$, such that $\mathbf{A}_j^T \mathbf{A}_j - \mathbf{E}_j = \mathbf{I}$, we introduce a network defined on $\Omega(\mathbf{b}_j)$ by:

$$\begin{aligned} \mathcal{N}(W, Z) :&\to x_j = \mathbf{z}_j^T \mathbf{o}_j = \mathbf{z}_j^T (\mathbf{A}_j^T \mathbf{A}_j - \mathbf{E}_j) \mathbf{o}_j = (\mathbf{A}_j \mathbf{z}_j)^T (\mathbf{A}_j \mathbf{o}_j) - \mathbf{z}_j^T \mathbf{E}_j \mathbf{o}_j \\ &= (\mathbf{b}_j^T \mathbf{A}_j - [\mathbf{z}_j(\mathbf{b}_j)]^T \mathbf{E}_j) \mathbf{o}_j :\to \mathcal{N}(W, B), \end{aligned} \tag{8}$$

where $x_j$ and $\mathbf{o}_j$ are defined in Eq. (1), $B$ denotes the architecture variables in the network $\mathcal{N}(W, B)$, i.e., $B = \{\mathbf{b}_j\}_{j=2}^n$, and $\mathbf{z}_j(\mathbf{b}_j)$ is deemed as a function of $\mathbf{b}_j$ in $\mathcal{N}(W, B)$. The two arrows denote how $x_j$ is produced in the networks $\mathcal{N}(W, Z)$ and $\mathcal{N}(W, B)$, respectively. We note that $\mathcal{N}(W, B)$ and $\mathcal{N}(W, Z)$ have the same propagation for node $x_j$, $\forall\, 1 < j \leq n$ when $\mathbf{b}_j = \mathbf{A}_j \mathbf{z}_j$.

Supposing there is only one architecture variable (i.e., $n = 2$), we remove the index $j$ for simplicity. We consider that $\mathbf{A}$ satisfies the restricted isometry property (RIP) [6] with the constant $\delta_{2s}$, which is the smallest $\delta \in (0, 1)$ such that $(1 - \delta)\|\mathbf{z}\|_2^2 \leq \|\mathbf{A}\mathbf{z}\|_2^2 \leq (1 + \delta)\|\mathbf{z}\|_2^2$ for all $2s$-sparse $\mathbf{z}$. Then for any $\mathbf{b} \in \Omega(\mathbf{b})$, there is at most one $s$-sparse $\mathbf{z} \in \Omega(\mathbf{z})$ that satisfies $\mathbf{A}\mathbf{z} = \mathbf{b}$. When $\delta$ meets a more strict condition, an exact $s$-sparse recovery of $\mathbf{z}$ is guaranteed via $\ell_1$-norm minimization [6]. Here we simply assume that the solution derived by $\mathbf{z}^* = \text{argmin}_{\mathbf{z}} \frac{1}{2}\|\mathbf{A}\mathbf{z} - \mathbf{b}\|_2^2 + \lambda\|\mathbf{z}\|_1$ is $s$-sparse and satisfies $\mathbf{A}\mathbf{z}^* = \mathbf{b}$, which is possible with $\lambda$ small enough. Then we have the following proposition:

**Proposition 1.** *Assume that $\mathbf{A}$ satisfies the RIP with its constant $\delta_{2s}$ and the exact $s$-sparse solution $\mathbf{z}^*$ can be recovered by $\text{argmin}_{\mathbf{z}} \frac{1}{2}\|\mathbf{A}\mathbf{z} - \mathbf{b}\|_2^2 + \lambda\|\mathbf{z}\|_1$ and satisfies $\mathbf{A}\mathbf{z}^* = \mathbf{b}$. Then we have that $\mathbf{z}^*$ is the optimal solution of the network $\mathcal{N}(W, \mathbf{z})$ if and only if $\mathbf{b}^* = \mathbf{A}\mathbf{z}^*$ is the optimal solution of the network $\mathcal{N}(W, \mathbf{b})$.*

*Proof.* For sufficiency, suppose that there exists another $s$-sparse $\mathbf{z}' \neq \mathbf{z}^*$, such that $\mathcal{L}(\mathcal{N}(W, \mathbf{z}')) < \mathcal{L}(\mathcal{N}(W, \mathbf{z}^*))$. Then $\mathbf{b}' = \mathbf{A}\mathbf{z}' \neq \mathbf{b}^*$ due to the RIP of $\mathbf{A}$. Because the two networks $\mathcal{N}(W, \mathbf{b})$ and $\mathcal{N}(W, \mathbf{z})$ have the same propagation by Eq. (8), we have $\mathcal{L}(\mathcal{N}(W, \mathbf{b}')) = \mathcal{L}(\mathcal{N}(W, \mathbf{z}')) < \mathcal{L}(\mathcal{N}(W, \mathbf{z}^*)) = \mathcal{L}(\mathcal{N}(W, \mathbf{b}^*))$, which is in conflict with the fact that $\mathbf{b}^* = \mathbf{A}\mathbf{z}^*$ is the optimal solution of $\mathcal{N}(W, \mathbf{b})$. The necessity can be derived in the same way. Suppose that there exists another $\mathbf{b}' \neq \mathbf{b}^*$, such that $\mathcal{L}(\mathcal{N}(W, \mathbf{b}')) < \mathcal{L}(\mathcal{N}(W, \mathbf{b}^*))$. Then $\mathbf{z}' = \text{argmin}_{\mathbf{z}} \frac{1}{2}\|\mathbf{A}\mathbf{z} - \mathbf{b}'\|_2^2 + \lambda\|\mathbf{z}\|_1 \neq \mathbf{z}^*$. We have $\mathcal{L}(\mathcal{N}(W, \mathbf{z}')) = \mathcal{L}(\mathcal{N}(W, \mathbf{b}')) < \mathcal{L}(\mathcal{N}(W, \mathbf{b}^*)) = \mathcal{L}(\mathcal{N}(W, \mathbf{z}^*))$, which is in conflict with the fact that $\mathbf{z}^*$ is the optimal solution of $\mathcal{N}(W, \mathbf{z})$, and concludes the proof. $\square$

**Algorithm 1** Two-stage ISTA-NAS (for search only)

---

**Input:** Initialize the network weights $W$ of the whole super-net $\mathcal{N}(W, B)$ and architecture variables $\mathbf{b}_j \in \mathbb{R}^{m_j}$ for each intermediate node $1 < j \leq n$. Sample $\mathbf{A}_j \in \mathbb{R}^{m_j \times (j-1)K}, \forall 1 < j \leq n$.
1: **while** *not converged* **do**
2:　　Recover $\mathbf{z}$ by solving Eq. (9) with ISTA. Keep the top-$s$ strongest magnitudes and set other dimensions as zeros. The support set $\mathcal{S}(\mathbf{z}) = \{i | \mathbf{z}(i) \neq 0\}$;
3:　　Derive a sub-graph $\mathcal{N}(W_{(\mathcal{S})}, B)$ of the super-net by only propagating the dimensions in $\mathcal{S}$;
4:　　Update network weights $W_{(\mathcal{S})}$ by descending $\nabla_{W_{(\mathcal{S})}} \mathcal{L}_{train}(\mathcal{N}(W_{(\mathcal{S})}, B))$;
5:　　Update architecture variables $\mathbf{b}$ by descending $\nabla_{\mathbf{b}} \mathcal{L}_{val}(\mathcal{N}(W_{(\mathcal{S})}, B))$;
6: **end while**
**Output:** Produce a sparse architecture for evaluation according to the final $\mathcal{S}(\mathbf{z})$.

---

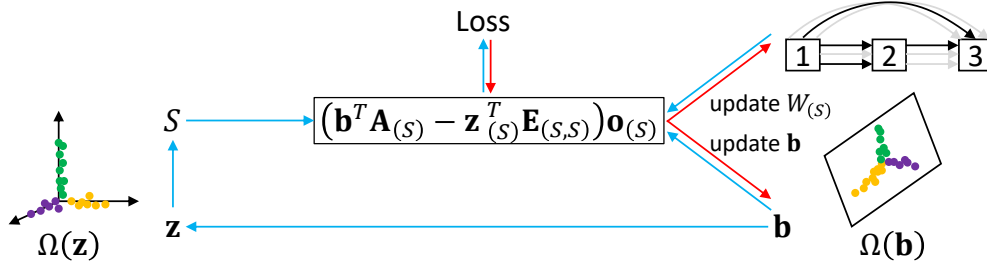

Figure 1: An illustration of our search process. The blue arrows denote the forward propagation, while the red ones are backward updates. $\Omega(\mathbf{z})$ and $\Omega(\mathbf{b})$ are the original and the compressed spaces, respectively. $\mathbf{z}$ lies near the axises because it is sparse. The black arrows in the network represent the corresponding connections that are active in the current $\mathcal{S}$, while the gray ones do not belong to $\mathcal{S}$.

It is shown that a proper measurement matrix $\mathbf{A}$ builds the relation between the original and the compressed spaces. The optimal solution in $\Omega(\mathbf{z})$ can be searched by optimization in $\Omega(\mathbf{b})$. Thus we have our problem formulated as:

$$\mathbf{z}_j = \underset{\mathbf{z}}{\arg\min} \frac{1}{2} \|\mathbf{A}_j \mathbf{z} - \mathbf{b}_j\|_2^2 + \lambda \|\mathbf{z}\|_1, \quad 1 < j \leq n, \tag{9}$$

$$\begin{cases} B^* = \underset{B}{\arg\min} \, \mathcal{L}_{val}(\mathcal{N}(W^*, B)), \\ W^* = \underset{W}{\arg\min} \, \mathcal{L}_{train}(\mathcal{N}(W, B)), \end{cases} \tag{10}$$

where $B = \{\mathbf{b}_j\}_{j=2}^n$ is the trainable variables in the network $\mathcal{N}(W, B)$. $B$ and $W$ are optimized by the differentiable NAS without the sparsity constraint Eq. (4). $Z = \{\mathbf{z}_j\}_{j=2}^n$ is recovered by the sparse coding problem in Eq. (9), which inherently produces a sparse architecture.

### 3.4 Two-stage ISTA-NAS

The benefit of introducing sparse coding into NAS is that we can utilize the sparsity for more efficient search. In implementation, the solution of Eq. (9) may not lie in $\Omega(\mathbf{z})$ and strictly satisfy the sparsity constraint. We keep the top-$s$ strongest magnitudes and set other dimensions as zeros. Let $\mathcal{S}(\mathbf{z})$ denote the support set of $\mathbf{z}$, *i.e.*, $\mathcal{S}(\mathbf{z}) = \{i | \mathbf{z}(i) \neq 0\}$ and $|\mathcal{S}| = s$, where $i$ is the $i$-th dimension of $\mathbf{z}$ and $s$ is the sparseness. Then the network propagation in Eq. (8) is equivalent to:

$$x = \mathbf{z}^T \mathbf{o} = \mathbf{z}_{(\mathcal{S})}^T \mathbf{o}_{(\mathcal{S})} = \mathbf{z}_{(\mathcal{S})}^T \left( \mathbf{A}_{(\mathcal{S})}^T \mathbf{A}_{(\mathcal{S})} - \mathbf{E}_{(\mathcal{S},\mathcal{S})} \right) \mathbf{o}_{(\mathcal{S})} = \left( \mathbf{b}^T \mathbf{A}_{(\mathcal{S})} - \mathbf{z}_{(\mathcal{S})}^T \mathbf{E}_{(\mathcal{S},\mathcal{S})} \right) \mathbf{o}_{(\mathcal{S})}, \tag{11}$$

where $\mathbf{z}_{(\mathcal{S})}$ denotes the elements of $\mathbf{z}$ indexed by $\mathcal{S}$, $\mathbf{A}_{(\mathcal{S})}$ denotes the columns of $\mathbf{A}$ indexed by $\mathcal{S}$, and $\mathbf{E}_{(\mathcal{S},\mathcal{S})}$ denotes the rows and columns of $\mathbf{E}$ indexed by $\mathcal{S}$.

We now introduce our two-stage ISTA-NAS outlined in Algorithm 1. The two-stage pipeline is consistent with current studies [30, 48, 9, 49]. A super-net with the whole DAG is constructed for search, after which the searched sparse architecture is for evaluation with a larger depth and width. We first initialize the network weights $W$ and architecture variables $B$ of the super-net. The measurement matrices $\mathbf{A}_j$ used to construct the relations between the original and the compressed

---
**Algorithm 2** One-stage ISTA-NAS (for both search and evaluation)
---
**Input:** Initialize $\mathcal{N}(W, B)$ with depth, width, and batch size in the target-net setting. $\gamma$ and $\beta$ of BN layers in all candidate operations are frozen and initialized as 1 and 0. $search\_flag := True$.
 1: **while** *not converged* **do**
 2:   **if** $search\_flag$ **then**
 3:     Perform the Line 2 and Line 3 of Algorithm 1; $\mathbf{z}^{new} := \mathbf{z}$;
 4:   **end if**
 5:   **if** $search\_flag$ **and** $\|\mathbf{z}^{new} - \mathbf{z}^{old}\| \leq \epsilon$ **then**
 6:     $\gamma.requires\_grad := True$; $\beta.requires\_grad := True$; $search\_flag := False$;
 7:   **end if**
 8:   Update network weights $W_{(\mathcal{S})}$ by descending $\nabla_{W_{(\mathcal{S})}}\mathcal{L}_{train}(\mathcal{N}(W_{(\mathcal{S})}, B))$;
 9:   **if** $search\_flag$ **then**
10:     Update architecture variables $\mathbf{b}$ by descending $\nabla_{\mathbf{b}}\mathcal{L}_{train}(\mathcal{N}(W_{(\mathcal{S})}, B))$; $\mathbf{z}^{old} := \mathbf{z}^{new}$;
11:   **end if**
12: **end while**
13: Update the parameters of BN layers by Eq. (12);
**Output:** Produce a sparse architecture and its optimized parameters.
---

spaces for each node are sampled and fixed. At each iteration of the algorithm, $\mathbf{b}_j$ corresponds to a sparse $\mathbf{z}_j$ by solving Eq. (9) with ISTA. Then we keep the top-$s$ strongest magnitudes of $\mathbf{z}_j$ and derive its support set $\mathcal{S}$, which indicates a sparse architecture. We only propagate the connections in $\mathcal{S}$ by Eq. (11), and then updates $W_{(\mathcal{S})}$ and $\mathbf{b}$, respectively. The optimization of Eq. (10) is performed in an alternate way, which is the same as the widely-used bi-level scheme in current differentiable NAS methods [30]. The difference lies in that $\mathcal{S}$ dynamically changes in our search, so the sparsity constraint is inherently satisfied at each update. We do not need a projection onto the target-net constraint by post-processing. Besides, the network $\mathcal{N}(W_{(\mathcal{S})}, B)$ in our search is sparse and more efficient to train. An illustration of how we perform our search process is shown in Figure 1.

### 3.5 One-stage ISTA-NAS

Since the super-net of ISTA-NAS is as sparse as the target-net, our method is friendly to memory and is able to adopt the larger depth and width in target-net. If we can use one network for both search and evaluation under the target-net settings, the gap of depth, width, and even batch size in two-stage methods will be eliminated. To this end, we develop a one-stage method, where search and evaluation are finished in a single run, after which we get both architecture and its optimized parameters.

In Eq. (11), the architecture variables $\mathbf{b}$ and matrix $\mathbf{A}$ decide the coefficient of each connection for differentiable search. We need to combine these coefficients with network weights $W$ as the final optimized parameters. Considering the parameterized operations for search, such as $3 \times 3$ separable convolution, $5 \times 5$ dilation convolution, end up with a batch normalization (BN) layer, we also append BN layers after non-parametric operations, including identity and all pooling operations.

The one-stage ISTA-NAS is outlined in Algorithm 2. At the beginning, the weight and bias of BN layers, *i.e.,* $\gamma$ and $\beta$, are frozen and initialized as 1 and 0, respectively, so they will not affect the search process. Different from the two-stage method, $W_{(\mathcal{S})}$ and $\mathbf{b}$ share the same loss using the full train set (no validation set is split out). We introduce a termination condition, such that when $\mathbf{z}$ of two neighboring iterations are close, *i.e.,* $\|\mathbf{z}^{new} - \mathbf{z}^{old}\| \leq \epsilon$, we stop the optimization of $\mathbf{b}$. Then $\mathbf{z}$ and $\mathcal{S}$ no longer change and thus a sparse architecture is searched and fixed. The coefficient of each connection does not change accordingly. The weight $\gamma$ and bias $\beta$ of BN layers are now enabled to optimize. When training finishes, the coefficients are absorbed into $\gamma$, $\beta \in \mathbb{R}^s$ as[2]:

$$\hat{\gamma} = \left(\mathbf{b}^T\mathbf{A}_{(\mathcal{S})} - \mathbf{z}_{(\mathcal{S})}^T\mathbf{E}_{(\mathcal{S},\mathcal{S})}\right) \circ \gamma; \quad \hat{\beta} = \left(\mathbf{b}^T\mathbf{A}_{(\mathcal{S})} - \mathbf{z}_{(\mathcal{S})}^T\mathbf{E}_{(\mathcal{S},\mathcal{S})}\right) \circ \beta; \quad (12)$$

where $\circ$ is the element-wise multiplication, and $\hat{\gamma}$, $\hat{\beta} \in \mathbb{R}^s$ are updated parameters that keep the trained network accuracy unchanged. In this way, we get the searched architecture and its optimized parameters in a single run of Algorithm 2. No further training is necessary.

| | Bs. | Mem. | Search Cost |
|---|---|---|---|
| DARTS (1st order) | 64 | 9.1 G | 0.70 day |
| PC-DARTS | 256 | 11.6 G | 0.14 day |
| ISTA-NAS | 64 | 1.9 G | 0.15 day |
| ISTA-NAS | 256 | 5.5 G | 0.05 day |
| ISTA-NAS | 512 | 10.5 G | 0.03 day |

Table 1: Search cost of DARTS, PC-DARTS, and two-stage ISTA-NAS. "Bs." and "Mem." denote the batchsize and its corresponding memory consumption. Search cost is tested on a GTX 1080Ti GPU.

| | Kendall $\tau$ |
|---|---|
| DARTS (1st order) | $-0.36$ |
| PC-DARTS | $-0.21$ |
| Two-stage ISTA-NAS | $0.43$ |
| One-stage ISTA-NAS | $0.57$ |

Table 2: Kendall $\tau$ correlation of search and evaluation performances in DARTS, PC-DARTS, and ISTA-NAS. The one-stage ISTA-NAS is measured by its converged accuracy and retraining without the optimization of $\mathbf{b}$.

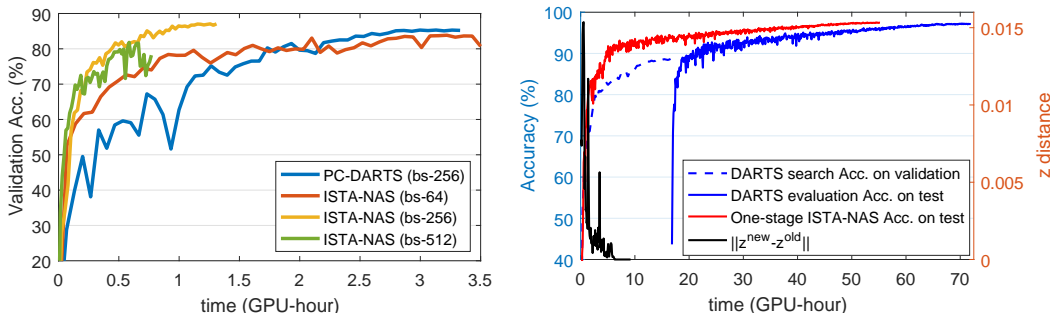

Figure 2: (left) The super-net accuracies of search in PC-DARTS and two-stage ISTA-NAS; (right) The search and evaluation processes of two-stage DARTS and one-stage ISTA-NAS.

## 4 Experiments

We analyze the improved efficiency and correlation of the two-stage and one-stage ISTA-NAS, and then compare our search results on CIFAR-10 and ImageNet with state-of-the-art methods. **All searched architectures are visualized in the supplementary material**.

### 4.1 Implementation Details

Our setting of search and evaluation is consistent with the convention in current studies [30, 48, 9, 49]. Please see the full description of our implementation details in the supplementary material. For our two-stage ISTA-NAS, the super-net is composed of 6 normals cells and 2 reduction cells. Each cell has 6 nodes. The first two nodes are input nodes output from the previous two cells. As convention, each intermediate node keeps two connections after search, so the sparseness $s_j = 2$ in our method. We adopt the Adam optimizer for $\mathbf{b}_j$ and SGD for network weights $W$. We use the released tool MOSEK with CVX [17] to efficiently solve Eq. (9). For our single-stage ISTA-NAS, we adopt the target-net settings of the evaluation stage. The network is stacked by 18 normal cells and 2 reduction cells. Other details are described in the supplementary material.

### 4.2 Efficiency and Correlation Improved by ISTA-NAS

ISTA-NAS inherently satisfies the sparsity constraint at each update, so significantly improves the search efficiency. As shown in Table 1, we re-implement DARTS and PC-DARTS using their released codes on CIFAR-10. When our two-stage ISTA-NAS uses a batchsize of 64, the search cost is on par with PC-DARTS, which uses a batchsize of 256 and consumes much more GPU memory. Two-stage ISTA-NAS supports a batchsize of 512, which leads to a search cost of only 0.03 GPU-day, about 20 times faster than DARTS. Their search processes are depicted in Figure 2 (left). The learning rates are also enlarged by the same times as batchsize. It is shown that ISTA-NAS with a batchsize of 256 has the most stable search accuracy on validation. Thus we adopt the 256 batchsize and 0.1 learning rate of $W$ for our two-stage search result on CIFAR-10. As for the one-stage ISTA-NAS, its accuracy on test is shown in Figure 2 (right). We also visualize the distance of $\mathbf{z}_j$ of two neighboring iterations averaged by all intermediate nodes $j$. It is observed that the distance of neighboring $\mathbf{z}$ decays fast. In about 10 GPU-hour, the termination condition is satisfied for all intermediate nodes, after which only network weights in the searched architecture get optimized. We have the architecture searched and optimized in a single run, which saves the search cost compared with the two-stage DARTS.

| Methods | Test Error (%) | Params (M) | Cost (GPU-day) search | Cost (GPU-day) eval. | Search Method |
|---|---|---|---|---|---|
| DenseNet-BC [22] | 3.46 | 25.6 | - | - | manual |
| NASNet-A + cutout [61] | 2.65 | 3.3 | 1800 | 3.2 | RL |
| ENAS + cutout [39] | 2.89 | 4.6 | 0.5 | 3.2 | RL |
| AmoebaNet-B +cutout [40] | 2.55±0.05 | 2.8 | 3150 | - | evolution |
| NAONet-WS [31] | 3.53 | 3.1 | 0.4 | - | NAO |
| DARTS (2nd order) + cutout [30] | 2.76±0.09 | 3.3 | 4.0 | 2.3 | gradient |
| SNAS (moderate) + cutout [48] | 2.85±0.02 | 2.8 | 1.5 | 2.2 | gradient |
| P-DARTS+cutout [9] | 2.50 | 3.4 | 0.3 | 2.9 | gradient |
| NASP + cutout [53] | 2.83±0.09 | 3.3 | 0.1 | - | gradient |
| PC-DARTS + cutout [49] | 2.57±0.07 | 3.6 | 0.1 | 3.1 | gradient |
| Two-stage ISTA-NAS + cutout | 2.54±0.05 | 3.32 | **0.05** | 2.0 | gradient |
| One-stage ISTA-NAS + cutout | **2.36**±0.06 | 3.37 | 2.3 | | gradient |

Table 3: Search results on CIFAR-10 and comparison with state-of-the-art methods. Cost is tested on a GTX 1080Ti GPU. The evaluation cost is calculated by us with their searched architectures in the same experimental settings. The cost of one-stage ISTA-NAS is the time spent in a single run.

| Methods | Test Err. (%) top-1 | Test Err. (%) top-5 | Params (M) | Flops (M) | Cost (GPU-day) search | Cost (GPU-day) eval. | Search Method |
|---|---|---|---|---|---|---|---|
| Inception-v1 [43] | 30.2 | 10.1 | 6.6 | 1448 | - | - | manual |
| MobileNet [20] | 29.4 | 10.5 | 4.2 | 569 | - | - | manual |
| ShuffleNet 2× (v2) [32] | 25.1 | - | ∼5 | 591 | - | - | manual |
| NASNet-A [61] | 26.0 | 8.4 | 5.3 | 564 | 1800 | - | RL |
| MnasNet-92 [44] | 25.2 | 8.0 | 4.4 | 388 | - | - | RL |
| AmoebaNet-C [40] | 24.3 | 7.6 | 6.4 | 570 | 3150 | - | evolution |
| DARTS (2nd order) [30] | 26.7 | 8.7 | 4.7 | 574 | 4.0 | 3.6×8 | gradient |
| SNAS [48] | 27.3 | 9.2 | 4.3 | 522 | 1.5 | 3.3×8 | gradient |
| P-DARTS [9] | 24.4 | 7.4 | 4.9 | 557 | 0.3 | 3.6×8 | gradient |
| ProxylessNAS (ImgNet) [5] | 24.9 | 7.5 | 7.1 | 465 | 8.3 | - | gradient |
| PC-DARTS (ImgNet) [49] | 24.2 | 7.3 | 5.3 | 597 | 3.8 | 3.9×8 | gradient |
| One-stage ISTA-NAS (C-10) | 25.1 | 7.7 | 4.78 | 550 | 2.3 | 3.4×8 | gradient |
| One-stage ISTA-NAS (ImgNet) | **24.0** | **7.1** | 5.65 | 638 | 4.2×8 | | gradient |

Table 4: Search results on ImageNet and comparison with state-of-the-art methods. Cost is tested on eight GTX 1080Ti GPUs. "ImgNet" denotes it is directly searched on ImageNet. Otherwise, it is searched on CIFAR-10 and then transfered to ImageNet for evaluation.

We also test the correlation between search and evaluation by the Kendall $\tau$ metric [26], which ranges from $-1$ to 1 and evaluates the rank correlation of data pairs. If $\tau = -1$, the ranking order is reversed, while if the ranking is identical, $\tau$ will be 1. The full introduction of this metric is appended in the supplementary material. We implement DARTS, PC-DARTS and two-stage ISTA-NAS for 8 times on CIFAR-10 with different seeds and calculate their Kendall $\tau$ metrics based on the super-net accuracies in search and the target-net accuracies in evaluation. We measure the metric of one-stage ISTA-NAS using its converged accuracy and the one retrained for the same epochs without optimizing **b**. As shown in Table 2, both two-stage and one-stage ISTA-NAS have positive correlation scores, and the one-stage ISTA-NAS has the best correlation. The Kendall $\tau$ metric is still not close to 1, because accuracies on CIFAR-10 are not so distinguishable, and are easily affected by random noise.

### 4.3 Search Results on CIFAR-10

The search results of two-stage and one-stage ISTA-NAS on CIFAR-10 are shown in Table 3. We see that two-stage ISTA-NAS achieves a state-of-the-art error rate of 2.54% within only 0.05 GPU-day. The error rate is on par with P-DARTS, while the search cost is reduced to one sixth of the number in P-DARTS, and a half of that in PC-DARTS and NASP, which are the fastest approaches for DARTS-based search space before ISTA-NAS. The one-stage ISTA-NAS unifies the search and

evaluation in a single run of 2.3 GPU-day, which is smaller than the total cost of most two-stage methods. In our implementation of one-stage ISTA-NAS, after the termination condition is satisfied, the same epochs of training as the evaluation stage in two-stage ISTA-NAS is performed. So its search cost is usually larger than that of two-stage ISTA-NAS because of its direct search in the evaluation settings that have a larger depth and width. But the performance gets better due to its improved consistency brought by further reducing the gaps apart from sparseness, such as depth, width and training batchsize.

## 4.4 Search Results on ImageNet

We use one-stage ISTA-NAS for experiments on ImageNet. As shown in Table 4, the architecture searched on CIFAR-10 has a top-1 error rate of 25.1%, which is better than DARTS by more than 1.5% top-1 error rate. Its search cost on CIFAR-10 is larger than that of SNAS and P-DARTS because our search is unified with evaluation in one stage. When one-stage ISTA-NAS is directly performed on ImageNet, we have a top-1/5 error rates of 24.0%/7.1%, which slightly surpasses PC-DARTS, the best performance to date of DARTS-based search space on ImageNet as we know. Still, the total cost is lower than most two-stage methods. PC-DARTS and our method both directly search on ImageNet. However, the difference is that our search is performed on the full training set instead of a sampled subset as adopted in PC-DARTS. Consequently, our search results may enjoy a better generalization ability due to the sufficient access to the data space.

## 5 Conclusion

In this paper, we formulate the NAS problem as optimizing a sparse solution from a high-dimensional space spanned by all candidate connections, and introduce a more efficient and consistent search method, named ISTA-NAS. The differentiable search is performed on a compressed space and the sparse solution is recovered by ISTA. The sparsity constraint is inherently satisfied at each update, which makes the search more efficient and consistent with the architecture for evaluation. We further develop a one-stage method that unifies the search and evaluation in a single run. Experiments verify the improved efficiency and correlation of two-stage and one-stage ISTA-NAS. The searched architectures surpass the state-of-the-art performances on both CIFAR-10 and ImageNet.

## Broader Impact

Our work proposes a new perspective to formulate the NAS problem and develops two algorithms that have better efficiency and correlation. The positive impacts are obvious. First, better architectures may be searched for some problems, so the practical application of neural network to the corresponding area can be fostered. It benefits industry because the products or services with more satisfactory performance can be employed. Second, NAS has been a resource demanding task. Our method saves much memory and time cost, which is friendly to environment and easy to use for practitioners. Finally, automation is believed as a complementary tool to human experts. We think that a good NAS method could bring researchers expertise in understanding neural architectures.

## Acknowledgment

Z. Lin is supported by NSF China (grant no.s 61625301 and 61731018), Major Scientific Research Project of Zhejiang Lab (grant no.s 2019KB0AC01 and 2019KB0AB02), Beijing Academy of Artificial Intelligence, and Qualcomm.

## Footnotes

[2]$\gamma$ and $\beta$ are viewed as vectors in $\mathbb{R}^s$ formed by all active connections to the same node.

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
