[Supplementary Material]

# Supplementary Material of
# ISTA-NAS: Efficient and Consistent Neural Architecture Search by Sparse Coding

**Yibo Yang[1,2], Hongyang Li[2], Shan You[3], Fei Wang[3], Chen Qian[3], Zhouchen Lin[2,∗]**
[1]Center for Data Science, Academy for Advanced Interdisciplinary Studies, Peking University
[2]Key Laboratory of Machine Perception (MOE), School of EECS, Peking University
[3]SenseTime
{ibo, lhy_ustb, zlin}@pku.edu.cn, {youshan, wangfei, qianchen}@sensetime.com

## 1 Implementation Details

### 1.1 Datasets and Candidate Operations

We perform our experiments on both CIFAR-10 and ImageNet. The CIFAR-10 dataset has 60,000 colored images in 10 classes, with 50,000 images for training and 10,000 images for testing. The images are normalized by mean and standard deviation. As convention, we perform the data augmentation by padding each image 4 pixels filled with 0 on each side and then randomly cropping a $32 \times 32$ patch from each image or its horizontal flip. The ImageNet dataset contains 1.2 million training images, 50,000 validation images, and 100,000 test images in 1,000 classes. We adopt the standard data augmentation for training. A $224 \times 224$ crop is randomly sampled from the images or its horizontal flip. The images are normalized by mean and standard deviation. We report the single-crop top-1/5 error rates on the validation set in our experiments.

The candidate operations are in accordance with current studies [3, 4]. They are $3 \times 3$ and $5 \times 5$ separable convolution, $3 \times 3$ and $5 \times 5$ dilated separable convolution, $3 \times 3$ max and average pooling, and skip-connect. We do not use the zero operation since our methods do not rely on a post-processing process to derive the searched architecture.

### 1.2 Two-stage ISTA-NAS

The pipeline of our two-stage ISTA-NAS on CIFAR-10 is consistent with current two-stage methods [3, 4] for fair comparison. Concretely, the super-net for search is composed of 6 normal cells and 2 reduction cells, and has an initial number of channels of 16. Each cell has 6 nodes. The first 2 nodes are input nodes output from the previous two cells. The output of each cell is all intermediate nodes concatenated along the channel dimension. As convention, each intermediate node keeps two connections after search, so the sparseness $s_j = 2$ in our method. The training set is split into two equal parts, with one for network weights $W$, and the other as the validation set for architecture variables. We train the super-net for 50 epochs with a batchsize of 256 on a single GPU. We use SGD to optimize the network weights $W$ with a momentum of 0.9, a weight decay of $3 \times 10^{-4}$, and an initial learning rate of 0.2 annealed down to zero by a cosine scheduler. The architecture variables $\mathbf{b}_j$ are optimized by Adam on the validation set with a learning rate of $6 \times 10^{-4}$, a momentum of (0.5, 0.999), and a weight decay of $1 \times 10^{-3}$. We adopt the released tool MOSEK with CVX [1] to solve the sparse coding problem. The $\lambda$ in Eq. (9) is set as $1 \times 10^{-5}$. We run our method for 5 times and choose the architecture that has the best performance on validation as the searched one.

---

[∗]Corresponding author.

In evaluation, the target-net is composed of 18 normal cells and 2 reduction cells, and has 36 initial channels. We train the target-net for 600 epochs on the full training set. The batchsize is 96 and the commonly used enhancements, such as cutout, dropout and auxiliary head are used. We use the SGD optimizer with a momentum of 0.9, a weight decay of $3 \times 10^{-4}$, and an initial learning rate of 0.025 that is annealed down to zero by a cosine scheduler. We run the evaluation for 5 times with different seeds and report the mean error rate with its standard deviation on test set.

## 1.3   One-stage ISTA-NAS

The one-stage ISTA-NAS only uses one network and its implementation settings are in accordance with the evaluation stage of the two-stage ISTA-NAS.

For experiments on CIFAR-10, the network is stacked by 18 normals cells and 2 reduction cells with each cell covering all candidate connections. The initial number of channel is 36 and training batchsize is 96. The enhancements are also used accordingly. We run the Algorithm 2 with a fixed learning rate of 0.025 using the SGD optimizer. When the termination condition is satisfied for all intermediate nodes, the algorithm continues to run for 600 epochs with the learning rate annealed down to zero by a cosine scheduler. Finally, we re-evaluate the searched architecture for 4 times by running without the optimization of architecture variables $\mathbf{b}_j$, and report the mean and standard deviation of the error rates of these 5 results.

For experiments on ImageNet, the network starts with three convolution layers with a stride of 2 to reduce the resolution from $224 \times 224$ to $28 \times 28$. Then 12 normal cells and 2 reduction cells are stacked with the initial number of channels as 48. The training batchsize is 1,024. Enhancements including label smoothing and auxiliary head are used. The Adam optimizer for architecture variables is used with a learning rate of $6 \times 10^{-3}$, a momentum of (0.5, 0.999), and a weight decay of $1 \times 10^{-3}$. The SGD optimizer for network weights adopts a momentum of 0.9 and a weight decay of $3 \times 10^{-5}$. Its initial learning rate is 0.5. When the termination condition is satisfied for all intermediate nodes, we continue to train for 250 epochs with the learning rate annealed down to zero linearly. Different from [5], our search is performed on the full training set instead of a sampled subset. When training finishes, we report the converged top-1/5 error rates on the validation set.

## 2   Kendall Correlation

The Kendall correlation metric [2] is proposed to measure the ranking correlation of pairwise data. For data pairs $(x_i, y_i)$ and $(x_j, y_j)$, if $x_i < x_j$ and $y_i < y_j$ (or $x_i > x_j$ and $y_i > y_j$), then we call the pair $(i, j)$ is concordant. Otherwise it is disconcordant. Assuming that there are $N$ samples, we have the Kendall correlation metric calculated as:

$$\tau = \frac{\sum_{i<j} \text{sign}(x_i - x_j)\text{sign}(y_i - y_j)}{\mathbb{C}_N^2} \tag{1}$$

where the denominator $\mathbb{C}_N^2$ is the total number data pairs, and the numerator expresses the difference between the number of concordant pairs and that of disconcordant pairs. It shown that the Kendall metric is able to measure the ranking correlation, and $\tau$ ranges from -1 to 1, which implies the ranking orders of $\{x\}$ and $\{y\}$ change from being totally reversed to being identical.

## 3   Visualization of Architectures

We visualize the searched architectures of our methods. The two-stage ISTA-NAS on CIFAR-10 is shown in Figure 1. The one-stage ISTA-NAS on CIFAR-10 is shown in Figure 2. The one-stage ISTA-NAS on ImageNet is shown in Figure 3.

(a) normal cell          (b) reduction cell

Figure 1: Two-stage ISTA-NAS on CIFAR-10

(a) normal cell          (b) reduction cell

Figure 2: One-stage ISTA-NAS on CIFAR-10

(a) normal cell          (b) reduction cell

Figure 3: One-stage ISTA-NAS on ImageNet