[Reviews · NeurIPS 2020]

Review 1

Summary and Contributions: This paper raises the problem that previous DARTS-based methods have some gaps between the trained supernet and the derived architecture, due to the sparsity constraints are not satisfied with the architecture parameters. Instead of projecting the supernet onto the sparsity constraint, this paper directly optimizes in the equalized parameters space, where the architecture parameters satisfy the constraint at each step.

Strengths: 1. The idea is novel and efficient. 2. By performing the differentiable search on a compressed space, which is then recovered using sparse coding, the sparsity constraint is enforced at each step, which improves the correlation between the supernet and the derived architecture. 3. Also, the convergence speed is faster than previous methods. 4. The paper is well-organized and easy to understand.

Weaknesses: 1. It’s not clear to me that as the sparse constraint is enforced at the beginning, is it sufficient to explore the entire search space? Could the authors specify the proportion of the explored architectures? 2. The results on ImageNet are not very impressive as it is comparable but not surpass previous methods.

Correctness: Yes.

Clarity: Yes.

Relation to Prior Work: The following one-stage NAS should be cited and discussed: [1] Mei et al., AtomNAS: Fine-Grained End-to-End Neural Architecture Search, ICLR 2019. [2] Gao et al., MTL-NAS: Task-Agnostic Neural Architecture Search towards General-Purpose Multi-Task Learning. arXiv:2003.14058.

Reproducibility: Yes

Additional Feedback: I have read the authors' rebuttal and remain my initial recommendation.


Review 2

Summary and Contributions: This work proposes to explicitly enforce sparsity on the architecture parameters during search process in NAS. This is achieved by sampling a dictionary matrix A and map the sparse architecture parameters to a low-dimensional space B, and perform the search on B. Both supernet weight W, architecture B, and the sparse constraint are jointly optimized.

Strengths: 1. The sparsity of architecture parameters is explicitly enforced via sparse coding. The gap between supernet and derived architecture is minimized since the redundant edges are inactive. 2. The authors further propose a one-stage search, where the network weight is being updated based on converged architecture parameters, thus returning both the derived architecture and optimized weights.

Weaknesses: 1. I could not see a strong motivation for explicitly enforcing sparsity on architecture parameters. This is because there are already many works trying to decouple the dependency of evaluating sub-networks on the training of supernet (i.e., making the correlation higher). This means that we have ways to explicitly decouple the network evaluation with supernet training without adding a sparsity regularizaiton. 2. Properly understanding Table 2 requires more experiment details. As far as I know, weight-sharing methods require the BN to be re-calculated [1] to properly measure the Kendall correlation. Other works that can reduce the gap between supernet and sub-networks (e.g. [3]) or can make the edges activated to be sparse like GDAS [2] are not compared. Moreover, there seems no explanation in main content regarding Table 2. 3. The one-stage method proposed basically focusing on training network weights W after the training of architecture parameters is converged. However, similar idea can also be achieved in other differentiable NAS framework, where one can continue training the supernet weights after the architecture remains little change. For example, in GDAS, after the entropy of edges is well minimized, the sampled architecture will be close to determnistic, and one can keep training W to obtain the optimal weights. Moreover, other one-stage methods like [4] are not compared nor discussed. ======================== After reading the author's response, most of my concerns have been addressed. I choose to accept this submission. ======================== [1] Guo, Zichao, et al. "Single path one-shot neural architecture search with uniform sampling." ICLR 2020. [2] X. Dong and Y. Yang. Searching for a robust neural architecture in four gpu hours. CVPR 2019. [3] Bender, Gabriel, et al. "Understanding and simplifying one-shot architecture search." ICML2018. [4] Cai, Han, Chuang Gan, and Song Han. "Once for all: Train one network and specialize it for efficient deployment." ICLR 2020.

Correctness: The overall formualation and pipeline makes sense to me.

Clarity: The overall written is good.

Relation to Prior Work: Some of the existing one-stage NAS work is not discussed, as mentioned above.

Reproducibility: No

Additional Feedback:


Review 3

Summary and Contributions: -----post rebuttal--------- First of all, I appreciate the feedback from the authors. In the rebuttal, the authors claimed that "Aj (1 < j _x0014_ n) are sampled as fixed matrices". To me, it is really weird. For a sparse coding method, the base matrices should also be optimized. I did not see the reason to just sample and fixed the basis in this paper. Thus, I suspect the correctness of this method. I will lower my score to suggest to reject. ---------------------------- This paper proposed to formulate differentiable NAS as a sparse coding optimization problem. Specifically, the basic idea of this work is that the differentiable search is performed on a compressed space, thus, by introducing sparse coding technique, the sparsity constraint could be satisfied. The paper reports competitive performance on both CIFAR10 and ImageNet dataset with one-stage and two-stage pipelines.

Strengths: 1. The idea of this paper is nice. It is really natural to utilize sparse coding techniques to resolve the decoding problem of differentiable NAS problem. 2. The experimental results are also quite competitive.

Weaknesses: 1. The details of the model are not that clear. For example, the authors should clarify the meaning of the "arrow" symbol in Equation 8. And it also need to clarify how is A optimized (or derived) for each iteration in equation 9. 2. Reproducibility. Since a sparse coding part is involved in the optimization, the model cannot be optimized directly by a differential manner and mosek is used for the sparse coding part. Moreover, the paper did not mention how the model is implemented (PyTorch, TensorFlow or other framework). The code of the paper should be provided for reproducibility.

Correctness: Might be correct. The details of the proof for Proposition 1 should be provided. So far the proof for Proposition 1 is too short.

Clarity: Satisfied.

Relation to Prior Work: good

Reproducibility: Yes

Additional Feedback:


Review 4

Summary and Contributions: This paper proposes to use sparse coding to bridge the gap between search and evaluation, with the help of sparsity constraint. It further proposes a one-stage framework, which avoids the original architecture-level hyper-parameters gaps. The efficiency and effectiveness have been validated in experiments.

Strengths: 1. Using sparse coding to solve the gap issue in NAS is novel and promising. I like this idea. The formulation and notations are neat. 2. The one-stage framework makes the overall method unified. There are also a performance improvement in the one-stage framework. 3. The correlation analysis in Table 2 is very important for the overall idea. It successfully validate the previous motivation.

Weaknesses: This paper is good, but some aspects are still required improvements. 1. [Structure of this paper] The major contribution of this paper is sparse coding formulation, but not the one-stage framework. I suggest that the authors shorten the words on two-stage and one-stage ISTA-NAS. Moreover, as the one-stage method performs consistently better, the two-stage one seems less necessary. 2. [Experiments] It is a bit disappointing to find the authors only conduct experiments on NASNet search space. We all know that many constraints, e.g., fixed depth and width, and drawbacks exist in this cell-based search space, e.g., inference time. This idea is promising and deserves more comprehensive results on other search space. I strongly suggest the authors apply it in at least one chain-based search space, like ProxyLess search space. 3. There are NEW hyper-parameters introduced by the proposed method, e.g., \epsilon in Algorithm 2. The authors should show ablations on how to choose it.

Correctness: Yes.

Clarity: Yes. It is clear for me to understand the idea.

Relation to Prior Work: For the first contribution, it has clearly discussed. For the second contribution, i.e., one-stage NAS, differences from DSNAS should be clarified. In addition, to my best knowledge, the first one-stage NAS method is not DSNAS, but ONCE-FOR-ALL (Han et al.). The authors miss it.

Reproducibility: Yes

Additional Feedback: The major idea in this paper is promising and novel. But some aspects can still be improved. I will give a better rate if my above concerns well explained.

[Author Response · NeurIPS 2020]

We thank all reviewers for the valuable comments.

**1. To Reviewer #2**

**Sufficient to explore the entire search space?** We note that even keeping the inactive paths does not indicate that the
exploration of search space is extended. When the sparsity constraint is not enforced, the search process just optimizes
a larger super-net that incurs inconsistency with the finally derived architecture, instead of exploring a larger search
space. The proportion of explored architectures is decided by how many architectures are covered by the search process.
In order to analyze quantitatively, we run the search of DARTS and our ISTA-NAS on CIFAR-10 for three times, with
50 epochs for each run. We count how many different architectures are covered in the 50 epochs. For DARTS, we need
to perform the architecture derivation (filtering inactive paths) for each epoch to get its architecture. We find that the
average number of architectures for the three-times experiments of DARTS is 35.7, while the number of ISAT-NAS is
34.3. This shows that there is not a significant difference. We hope this result could resolve your concern.

**The results on ImageNet.** As shown in Tables 3 and 4, ISTA-NAS has higher accuracies than PC-DARTS on both
CIFAR-10 and ImageNet with less search cost. The superiority of accuracy on ImageNet is not so significant, maybe
because that PC-DARTS has been close to the upper limitation of performance on ImageNet in the DARTS-based space.

**Relations to AtomNAS and MTL-NAS.** Thanks for reminding us. We will cite and discuss the relations to AtomNAS
(ICLR 2020) and MTL-NAS (CVPR 2020) in the revised version.

**2. To Reviewer #3**

**Motivation.** We think that the decoupling method that you mentioned refers to sampling based methods, such as SPOS.
We need to point out that our method belongs to differentiable NAS methods, instead of sampling-based ones that could
have no architecture parameter (e.g. by the uniform sampling in SPOS). In the scope of differentiable NAS, without
enforcing sparsity, the super-net will be dense and incur inconsistency and redundant search cost. Please refer to lines
32 to 40 for our motivations. Besides, GDAS that minimizes the entropy of edges has similar motivations as ours.

**Kendall calculation.** Note that the methods of the chain-based space, such as SPOS, need to re-calculate BN layers
for measuring the accuracy of the sub-network extracted from a super-net. But our method belongs to the cell-based
search space. We just measure the correlation between the accuracies of the super-net after search and the target-net
after retraining, so do not have the BN problem. The explanation of Table 2 is shown in lines 246 to 255 of our paper.

**Comparison with GDAS and One-shot.** We compare with One-shot (Bender, et al. ICML 2018) and GDAS. On
CIFAR-10, GDAS has an error rate of 2.93% with 0.21 GPU-day, while ISTA-NAS has an error rate of 2.54% with
0.05 GPU day. On ImageNet, the top-1 accuracies of One-shot, GDAS, and ISTA-NAS are 75.2%, 74.0%, and 76%,
respectively. Besides, GDAS did not have a one-stage method and direct search on ImageNet in their paper.

**Discussion of Once for All (OFA).** OFA searches in the chain-based space so its result could not be directly compared
with ours of the cell-based space. We will cite and discuss it in detail in the revised version.

**3. To Reviewer #6**

**Clear details.** Thanks for your suggestions. The arrows in Eq. (8) just denote how $x_j$ is computed in the network
$\mathcal{N}(W, Z)$ and how to get $\mathcal{N}(W, B)$. We will make the description clearer in the revised version. As shown in "Input"
of Algorithm 1, $\mathbf{A}_j$ $(1 < j \leq n)$ are sampled as fixed matrices. They construct the relations between the original
space and the compressed space, and are not learnable in Eq. (9). The proof of Proposition 1 does not unfold some
easily-derived steps due to the limited page space. We will include the complete proof in the revised version.

**Implementations.** The implementation details are described in the supplementary material. The code is based on
PyTorch. But the sparse coding part is achieved on CPU by a Python interface of CVX using the MOSEK solver. So
the whole framework is still differentiable for training. Code address will be released.

**4. To Reviewer #7**

**Structure.** Thanks for your suggestion. We start with the two-stage method because it offers the basis of the one-stage
method. The one-stage method enjoys better consistency but performs search directly in the evaluation settings, which
consumes more time in general. So we keep both versions. We will consider your suggestion and refine the structure.

**Chain-based space.** We develop our method for the cell-based space because it has a higher dimension for intermediate
nodes, so suffers more than the chain-based space that mainly searches for the configurations of MBConv layers. But it
is a good suggestion to extend our method to chain-based space. Following the space of ProxylessNAS on ImageNet,
we get an architecture with 75.8% top-1 accuracy, which is a little better than ProxylessNAS (75.1%). It has less FLOPs
(410M) than our ISTA-NAS (cell-based) maybe due to the design of chain-based space. We think the result will be
better with more hyper-parameter tunning or if it can be combined with more advanced space, such as Once for all.

**Hyper-parameter.** As shown in Figure 2 (right), the difference of **z** in neighboring epochs decays fast. Even without
the termination condition, we observe that once the value is smaller than 1e-3, it will stay close to 0 and does not change
the architecture. So $\epsilon$ needs to be small, but no matter $\epsilon$ is 1e-3 or 1e-4, it has little effect on the search results.

**Discussion of Once for All.** Thanks for reminding us. We will cite and discuss it in detail in the revised version.

[Meta-Review · NeurIPS 2020]

Four knowledgeable reviewers support acceptance for the contributions. Reviewers find that i) using sparse coding to solve the gap issue in NAS is novel and promising. The formulation and notations are neat. ii) the one-stage framework makes the overall method unified. There is also a performance improvement in the one-stage framework. iii) the experimental results are also quite competitive. iv) Also, the convergence speed is faster than previous methods. V) the paper is well-organized and easy to understand. Therefore, I also recommend acceptance. However, please consider revising your paper to address all the concerns and comments from the reviewers. Specially, R6 asked to clarify the claim that Aj is sampled and fixed during training.